# Unsupervised Sound Separation Using Mixture Invariant Training

**Scott Wisdom**
Google Research
scottwisdom@google.com

**Efthymios Tzinis**[*]
UIUC
etzinis2@illinois.edu

**Hakan Erdogan**
Google Research
hakanerdogan@google.com

**Ron J. Weiss**
Google Research
ronw@google.com

**Kevin Wilson**
Google Research
kwwilson@google.com

**John R. Hershey**
Google Research
johnhershey@google.com

## Abstract

In recent years, rapid progress has been made on the problem of single-channel sound separation using supervised training of deep neural networks. In such supervised approaches, a model is trained to predict the component sources from synthetic mixtures created by adding up isolated ground-truth sources. Reliance on this synthetic training data is problematic because good performance depends upon the degree of match between the training data and real-world audio, especially in terms of the acoustic conditions and distribution of sources. The acoustic properties can be challenging to accurately simulate, and the distribution of sound types may be hard to replicate. In this paper, we propose a completely unsupervised method, mixture invariant training (MixIT), that requires only single-channel acoustic mixtures. In MixIT, training examples are constructed by mixing together existing mixtures, and the model separates them into a variable number of latent sources, such that the separated sources can be remixed to approximate the original mixtures. We show that MixIT can achieve competitive performance compared to supervised methods on speech separation. Using MixIT in a semi-supervised learning setting enables unsupervised domain adaptation and learning from large amounts of real-world data without ground-truth source waveforms. In particular, we significantly improve reverberant speech separation performance by incorporating reverberant mixtures, train a speech enhancement system from noisy mixtures, and improve universal sound separation by incorporating a large amount of in-the-wild data.

## 1 Introduction

Audio perception is fraught with a fundamental problem: individual sounds are convolved with unknown acoustic reverberation functions and mixed together at the acoustic sensor in a way that is impossible to disentangle without prior knowledge of the source characteristics. It is a hallmark of human hearing that we are able to hear the nuances of different sources, even when presented with a monaural mixture of sounds. In recent years significant progress has been made on extracting estimates of each source from single-channel recordings, using supervised deep learning methods. These techniques have been applied to important tasks such as speaker-independent enhancement (separation of speech from nonspeech interference) [19, 46] and speech separation (separation of speech from speech) [17, 20, 50]. The more general "universal sound separation" problem of separating arbitrary classes of sound from each other has also recently been addressed [21, 43].

---

[*]Work done during an internship at Google.

These approaches have used supervised training, in which ground-truth source waveforms are considered targets for various loss functions including mask-based deep clustering [17] and permutation invariant signal-level losses [20, 50]. Deep clustering is an embedding-based approach that implicitly represents the assignment of elements of a mixture, such as time-frequency bins of a spectrogram, to sources in a way that is independent of any ordering of the sources. In permutation invariant training [20, 50], the model explicitly outputs the signals in an arbitrary order, and the loss function finds the permutation of that order that best matches the estimated signals to the references, i.e. treating the problem as a set prediction task. In both cases the ground-truth signals are inherently part of the loss.

A major problem with supervised training for source separation is that it is not feasible to record both the mixture signal and the individual ground-truth source signals in an acoustic environment, because source recordings are contaminated by cross-talk. Therefore supervised training has relied on synthetic mixtures created by adding up isolated ground-truth sources, with or without a simulation of the acoustic environment. Although supervised training has been effective in training models that perform well on data that match the same distribution of mixtures, they fare poorly when there is mismatch in the distribution of sound types [31], or in acoustic conditions such as reverberation [30]. It is difficult to match the characteristics of a real dataset because the distribution of source types and room characteristics may be unknown and difficult to estimate, data of every source type in isolation may not be readily available, and accurately simulating realistic acoustics is challenging.

One approach to avoiding these difficulties is to use acoustic mixtures from the target domain, without references, directly in training. To that end, weakly supervised training has been proposed to substitute the strong labels of source references with another modality such as class labels, visual features, or spatial information. In [35] class labels were used as a substitute for signal-level losses. The spatial locations of individual sources, which can be inferred from multichannel audio, has also been used to guide learning of single-channel separation [42, 38, 10]. Visual input corresponding to each source has been used to supervise the extraction of the corresponding sources in [14], where the targets included mixtures of sources, and the mapping between source estimates and mixture references was given by the video correspondence. Because these approaches rely on multimodal training data containing extra input in the form of labels, video, or multichannel signals, they cannot be used in settings where only single-channel audio is available.

In this paper, we propose *mixture invariant training* (MixIT), a novel unsupervised training framework that requires only single-channel acoustic mixtures. This framework generalizes permutation invariant training (PIT) [50], in that the permutation used to match source estimates to source references is relaxed to allow summation over some of the sources. Instead of single-source references, MixIT uses mixtures from the target domain as references, and the input to the separation model is formed by summing together these mixtures to form a mixture of mixtures (MoM). The model is trained to separate this input into a variable number of latent sources, such that the separated sources can be remixed to approximate the original mixtures.

**Contributions**: (1) we propose the first purely unsupervised learning method that is effective for audio-only single-channel separation tasks such as speech separation and find that it can achieve competitive performance compared to supervised methods; (2) we provide extensive experiments with cross-domain adaptation to show the effectiveness of MixIT for domain adaptation to different reverberation characteristics in semi-supervised settings; (3) the proposed method opens up the use of a wider variety of data, such as training speech enhancement models from noisy mixtures by only using speech activity labels, or improving performance of universal sound separation models by training on large amounts of unlabeled, in-the-wild data.

## 2    Relation to previous work

Early separation approaches used hidden Markov models [36, 16, 23] and non-negative matrix factorization [39, 37] trained on isolated single-source data. These generative models incorporated the signal combination model into the likelihood function. The explicit generative construction enabled maximum likelihood inference and unsupervised adaptation [45]. However, the difficulty of discriminative training, restrictive modeling assumptions, and the need for approximation methods for tractable inference were liabilities. MixIT avoids these issues by performing self-supervised discriminative training of unrestricted deep networks, while still enabling unsupervised adaptation.

Discriminative source separation models generate synthetic mixtures from isolated sources which are also used as targets for training. Considering synthesis as part of the learning framework, such approaches can be described as self-supervised in that they start with single-source data. Early methods posed the problem in terms of time-frequency mask estimation, and considered restrictive cases such as speaker-dependent models, and class-specific separation, e.g. speech versus music [19], or noise [46]. However, more general speaker-independent speech separation [17, 50], and class-independent universal sound separation [21] are now addressed using class-independent methods such as deep clustering [17] and PIT [50]. These frameworks handle the output permutation problem caused by the lack of a unique source class for each output. Recent state-of-the-art models have shifted from mask-based recurrent networks to time-domain convolutional networks for speech separation [29], speech enhancement [21], and universal sound separation [21, 43] tasks.

MixIT follows this trend and uses a signal-level discriminative loss. The framework can be used with any architecture; in this paper we use a modern time-convolutional network. Unlike previous supervised approaches, MixIT can use a database of only mixtures as references, enabling training directly on target-domain mixtures for which ground-truth source signals cannot be obtained.

Similar to MixIT, Gao and Grauman [14] used MoMs as input, and summed over estimated sources to match target mixtures, using the *co-separation loss*. However, this loss does not identify correspondence between sources and mixtures, since that is established by the supervising video inputs, each of which is assumed to correspond to one source. In MixIT this is handled in an unsupervised manner, by finding the best correspondence between sums of sources and the reference mixtures without using other modalities, making our method the first fully unsupervised separation work using MoMs.

Also related is *adversarial unmix-and-remix* [18], which separates linear image mixtures in a GAN framework, with the discriminator operating on mixtures rather than single sources. Mixtures are separated, and the resulting sources are recombined to form new mixtures. Adversarial training encourages new mixtures to match the distribution of the original inputs. A cycle consistency loss is also used by separating and remixing the new mixtures. In contrast, MixIT avoids the difficulty of saddle-point optimization associated with GANs. An advantage of unmix-and-remix is that it is trained with only the original mixtures as input, while MixIT uses MoMs, relying on generalization to work on single mixtures. Unmix-and-remix was reported to work well on image mixtures, but failed on audio mixtures [18]. We show that MixIT works well on several audio tasks. However, unmix-and-remix is complementary, and could be combined with MixIT in future work.

Mixing inputs and outputs as in MixIT is reminiscent of MixUp regularization [52], which has been a useful component of recent techniques for semi-supervised classification [6, 5]. Our approach differs from these in that the sound separation problem is regression rather than classification, and thus training targets are not discrete labels, but waveforms in the same domain as the model inputs. Moreover, our approach is unsupervised, whereas MixUp is a regularization for a supervised task.

In our experiments we explore adapting separation models to domains for which it is difficult to obtain reference source signals. Recent approaches to such unsupervised domain adaptation have used adversarial training to learn domain-invariant intermediate network activations [13, 8, 41], learn to translate synthetic inputs to the target domain [7], or train student and teacher models to predict consistent separated estimates from supervised and unsupervised mixtures [24]. In contrast, we take a semi-supervised learning approach and jointly train the same network using both supervised and unsupervised losses, without making explicit use of domain labels. A related approach was proposed for speech enhancement [3], inspired by prior work [26] in vision, which uses a self-supervised loss that requires a second uncorrelated realization of the same noisy input signal, obtained from a specialized mid-side microphone. In contrast, the proposed unsupervised loss only requires single-channel mixture recordings with minimal assumptions.

## 3   Method

We generalize the permutation invariant training framework to operate directly on unsupervised mixtures, as illustrated in Figure 1. Formally, a supervised separation dataset is comprised of pairs of input mixtures $x = \sum_{n=1}^{N} s_n$ and their constituent sources $\mathbf{s}_n \in \mathbb{R}^T$, where each mixture contains up to $N$ sources with $T$ time samples each. Without loss of generality, for the mixtures that contain only $N' < N$ sources we assume that $s_n = 0$ for $N' < n \leq N$. An unsupervised dataset only contains

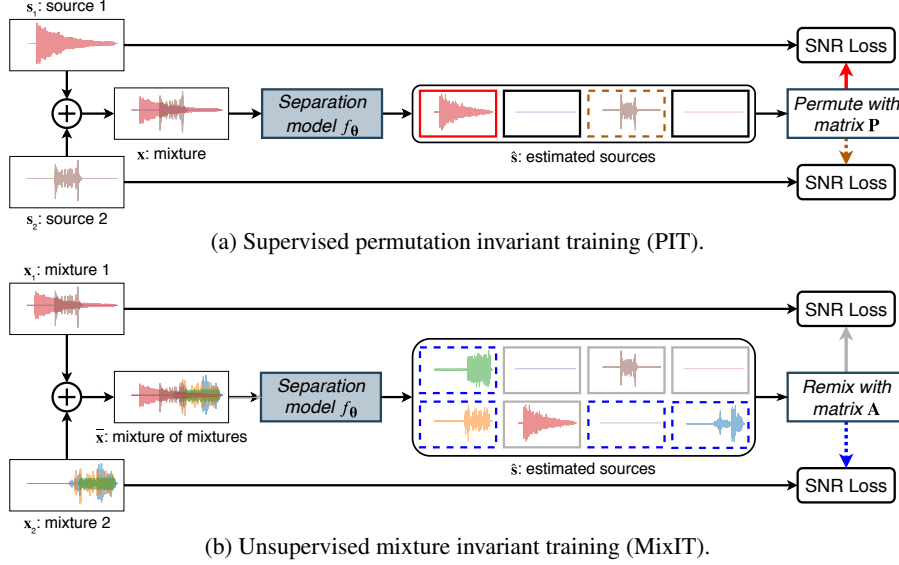

(a) Supervised permutation invariant training (PIT).

(b) Unsupervised mixture invariant training (MixIT).

Figure 1: Schematic comparing (a) PIT separating a two source mixture into up to four constituent sources to (b) MixIT separating a MoM into up to eight constituent sources. Arrow color indicates the best match between estimated sources and the ground-truth.

input mixtures without underlying reference sources. However, we assume that the maximum number of sources which may be present in the mixtures is known.

## 3.1 Permutation invariant training (PIT)

In the supervised case we are given a mixture $x$ and its corresponding sources $\mathbf{s}$ to train on. The input mixture $x$ is fed through a separation model $f_{\boldsymbol{\theta}}$ with parameters $\boldsymbol{\theta}$. The model predicts $M$ sources: $\hat{\mathbf{s}} = f_{\boldsymbol{\theta}}(x)$, where $M = N$ is the maximum number of sources co-existing in any given mixture drawn from the supervised dataset. Consequently, the supervised separation loss can be written as:

$$\mathcal{L}_{\text{PIT}}\left(\mathbf{s}, \hat{\mathbf{s}}\right) = \min_{\mathbf{P}} \sum_{m=1}^{M} \mathcal{L}\left(s_m, [\mathbf{P}\hat{\mathbf{s}}]_m\right), \tag{1}$$

where $\mathbf{P}$ is an $M \times M$ permutation matrix and $\mathcal{L}$ is a signal-level loss function such as negative signal-to-noise ratio (SNR). There is no predefined ordering of the source signals. Instead, the loss is computed using the permutation which gives the best match between ground-truth reference sources $\mathbf{s}$ and estimated sources $\hat{\mathbf{s}}$.

The signal-level loss function between a reference $y \in \mathbb{R}^T$ and estimate $\hat{y} \in \mathbb{R}^T$ from a model with trainable parameters $\boldsymbol{\theta}$ is the negative thresholded SNR:

$$\mathcal{L}(y, \hat{y}) = -10 \log_{10} \frac{\|y\|^2}{\|y - \hat{y}\|^2 + \tau \|y\|^2} = 10 \log_{10}\left(\|y - \hat{y}\|^2 + \tau \|y\|^2\right) \underbrace{-10 \log_{10} \|y\|^2}_{\text{const. w.r.t. } \boldsymbol{\theta}}, \tag{2}$$

where $\tau = 10^{-\text{SNR}_{\max}/10}$ acts as a soft threshold that clamps the loss at $\text{SNR}_{\max}$. This threshold prevents examples that are already well-separated from dominating the gradients within a training batch. We found $\text{SNR}_{\max} = 30$ dB to be a good value, as shown in Appendix C.

## 3.2 Mixture invariant training (MixIT)

The main limitation of PIT is that it requires knowledge of the ground truth source signals $\mathbf{s}$, and therefore cannot directly leverage unsupervised data where only mixtures $x$ are observed. MixIT overcomes this problem as follows. Consider two mixtures $x_1$ and $x_2$, each comprised of up to $N$ underlying sources (any number of mixtures can be used, but here we use two for simplicity). The mixtures are drawn at random without replacement from an unsupervised dataset and a MoM is

formed by adding them together: $\bar{x} = x_1 + x_2$. The separation model $f_{\boldsymbol{\theta}}$ takes $\bar{x}$ as input, and predicts $M \geq 2N$ source signals. In this way we make sure that the model is always capable of predicting enough sources for any $\bar{x}$. The unsupervised MixIT loss is computed between the estimated sources $\hat{\mathbf{s}}$ and the input mixtures $x_1$, $x_2$ as follows:

$$\mathcal{L}_{\mathrm{MixIT}}\left(x_1, x_2, \hat{\mathbf{s}}\right) = \min_{\mathbf{A}} \sum_{i=1}^{2} \mathcal{L}\left(x_i, [\mathbf{A}\hat{\mathbf{s}}]_i\right), \qquad (3)$$

where $\mathcal{L}$ is the same signal-level loss used in PIT (2) and the *mixing matrix* $\mathbf{A} \in \mathbb{B}^{2 \times M}$ is constrained to the set of $2 \times M$ binary matrices where each column sums to 1, i.e. the set of matrices which assign each source $\hat{s}_m$ to either $x_1$ or $x_2$. MixIT minimizes the total loss between mixtures $\mathbf{x}$ and remixed separated sources $\hat{\mathbf{x}} = \mathbf{A}\hat{\mathbf{s}}$ by choosing the best match between sources and mixtures, in a process analogous to PIT.

In practice, we optimize over $\mathbf{A}$ using an exhaustive $\mathcal{O}(2^M)$ search. The tasks considered in this paper only require $M$ up to 8, which we empirically found to not take a significant amount of time during training. To scale to larger values of $M$, it should be possible to perform this optimization more efficiently, but we defer this to future work.

There is an implicit assumption in MixIT that the sources are independent of each other in the original mixtures $x_1$ and $x_2$, in the sense that there is no information in the MoM $\bar{x}$ about which sources belong to which mixtures. The two mixtures $\mathbf{x} \in \mathbb{R}^{2 \times T}$ are assumed to result from mixing unknown sources $\mathbf{s}^* \in \mathbb{R}^{P \times T}$ using an unknown $2 \times P$ mixing matrix $\mathbf{A}^*$: $\mathbf{x} = \mathbf{A}^* \mathbf{s}^*$. If the network could infer which sources belong together in the references, and hence knew the mixing matrix $\mathbf{A}^*$ (up to a left permutation), then the $M$ source estimates $\hat{\mathbf{s}} \in \mathbb{R}^{M \times T}$ could minimize the loss (3) without cleanly separating all the sources (e.g., by under-separating). That is, for a known mixing matrix $\mathbf{A}^*$, the loss (3) could be minimized, for example, by the estimate $\hat{\mathbf{s}} = \mathbf{C}^+ \mathbf{A}^* \mathbf{s}^*$, with $\mathbf{C}^+$ the pseudoinverse of a $2 \times M$ mixing matrix $\mathbf{C}$ such that $\mathbf{C}\mathbf{C}^+ = \mathbf{I}$, at $\mathbf{A} = \mathbf{C}$, since $\mathbf{C}\hat{\mathbf{s}} = \mathbf{C}\mathbf{C}^+ \mathbf{A}^* \mathbf{s}^* = \mathbf{x}$. However, if the sources are independent, then the network cannot infer the mixing matrix that produced the reference mixtures. Nevertheless, the loss can be minimized with a single set of estimates, regardless of the mixing matrix $\mathbf{A}^*$, by separating all of the sources. That is, the estimated sources must be within a mixing matrix $\mathbf{B} \in \mathbb{B}^{P \times M}$ of the original sources, $\mathbf{s}^* = \mathbf{B}\hat{\mathbf{s}}$, so that (3) is minimized at $\mathbf{A} = \mathbf{A}^* \mathbf{B}$, for any $\mathbf{A}^*$. Hence, lack of knowledge about which sources belong to which mixtures encourages the network to separate as much as possible.

Note that when $M > P$, the network can produce more estimates than there are sources (i.e., over-separate). There is no penalty in (3) for over-separating the sources; in this work semi-supervised training helps with this, and future work will address methods to discourage over-separation in the fully unsupervised case.

### 3.3 Semi-supervised training

When trained on $M$ isolated sources, i.e. with full supervision, the MixIT loss is equivalent to PIT. Specifically, input mixtures $x_i$ are replaced with ground-truth reference sources $s_m$ and the mixing matrix $\mathbf{A}$ becomes an $M \times M$ permutation matrix $\mathbf{P}$. This makes it straightforward to combine both losses to perform semi-supervised learning. In essence, each training batch contains $p\%$ supervised data, for which we use the PIT loss (1), and the remaining contains unsupervised mixtures, for which we do not know their constituent sources and use the MixIT loss (3).

## 4   Experiments

Our separation model $f_{\boldsymbol{\theta}}$ consists of a learnable convolutional basis transform that produces mixture basis coefficients. These are processed by an improved time-domain convolutional network (TDCN++) [21], similar to Conv-TasNet [29]. This network predicts $M$ masks with the same size as the basis coefficients. Each mask contains values between 0 and 1. These masks are multiplied element-wise with the basis coefficients, a decoder matrix is applied to transform to masked coefficients to time-domain frames, and $M$ separated waveforms are produced by overlapping and adding these time-domain frames (also known as a transposed 1D convolutional layer). A mixture consistency projection layer [49] is applied to constrain separated sources to add up to the input

mixture. The architecture is described in detail in Appendix A. All models are trained on 4 Google Cloud TPUs (16 chips) with the Adam optimizer [22], a batch size of 256, and learning rate of $10^{-3}$.

Separation performance is measured using scale-invariant signal-to-noise ratio (SI-SNR) [25]. SI-SNR measures the fidelity between a signal $y$ and its estimate $\hat{y}$ within an arbitrary scale factor:

$$\text{SI-SNR}(y, \hat{y}) = 10 \log_{10} \frac{\|\alpha y\|^2}{\|\alpha y - \hat{y}\|^2}, \tag{4}$$

where $\alpha = \text{argmin}_a \|ay - \hat{y}\|^2 = y^T \hat{y} / \|y\|^2$. Generally we measure SI-SNR improvement (SI-SNRi), which is the difference between the SI-SNR of each source estimate after processing, and the SI-SNR obtained using the input mixture as the estimate for each source. For mixtures that contain only a single source, SI-SNRi is not meaningful, because the mixture SI-SNR is infinite. In this case, we measure the reconstruction performance using single-source absolute SI-SNR (SS). In real-world separation tasks, mixtures can contain a variable number of sources, or fewer sources than are produced by the separation model. To handle these cases during evaluation, we compute a multi-source SI-SNRi (MSi) metric by zero-padding references to $M$ sources, aligning them to the separated sources with a permutation maximizing SI-SNR, and averaging the resulting SI-SNRi over non-zero references. We provide audio demos for all tasks [1], and code is available on GitHub [2].

## 4.1 Speech separation

For speech separation experiments, we use the WSJ0-2mix [17] and Libri2Mix [9] datasets, sampled at 8 kHz and 16 kHz. We also employ the reverberant spatialized version of WSJ0-2mix [44] and a reverberant version of Libri2Mix we created. Both datasets consist of utterances from male and female speakers drawn from either the Wall Street Journal (WSJ0) corpus or from LibriSpeech [33]. Reverberant versions are created by convolving utterances with room impulse responses generated by a room simulator employing the image method [4]. WSJ0-2mix provides 30 hours of training mixtures with individual source utterances drawn with replacement, and the train-360-clean split of Libri2Mix provides 364 hours of mixtures where source utterances are drawn without replacement.

For our experiment, we sweep the amount of supervised versus unsupervised data for both the anechoic and reverberant versions of WSJ0-2mix. The proportion $p$ of unsupervised data from the same domain is swept from 0% to 100% where supervised training uses PIT with the separation loss (2) between ground-truth references and separated sources, and unsupervised training only uses the mixtures using MixIT (3) with the same separation loss (2) between mixtures and remixed separated sources. In both cases, the input to the separation model is a mixture of two mixtures. For training, 3 second clips are used for WSJ0-2mix, and 10 second clips for Libri2Mix.

We try two variants of this task: mixtures that always contain two speakers (2-source) such that MoMs always contain four sources, and mixtures containing either one or two speakers (1-or-2-source) such that MoMs contain two to four sources. Note that the network always has four outputs. Evaluation always uses single mixtures of two sources. To determine whether unsupervised data can help with domain mismatch, we also consider using supervised data from a mismatched domain of reverberant mixtures. This mismatch simulates the realistic scenario faced by practitioners training sound separation systems, where real acoustic mixtures from a target domain are available without references and synthetic supervised data must be created to match the distribution of the real data. It is difficult to perfectly match the distribution of real data, so synthetic supervised data will inevitably have some mismatch to the target domain.

Results on single anechoic and reverberant 2-source mixtures are shown in Figure 2, and results on single-source inputs are in Appendix E. First, notice that reverberant data is more challenging to separate because reverberation smears out spectral energy over time, and thus all models achieve lower performance on reverberant data. Models trained on MoMs of 2-source mixtures tend to do less well compared to models trained on MoMs of 1-or-2-source mixture. One difference with the 1-or-2-source setup is that the model observes some inputs that have two sources, which matches the evaluation. Another difference is that as references, 1-source mixtures act as supervised examples.

Notice that for both anechoic and reverberant test data, even completely unsupervised training with MixIT on MoMs of 1-or-2-source mixtures (rightmost points) achieves performance on par with supervised training (leftmost points). For MoMs of 2-source mixtures, totally unsupervised MixIT performance is generally worse by up to 3 dB compared to fully or semi-supervised on anechoic

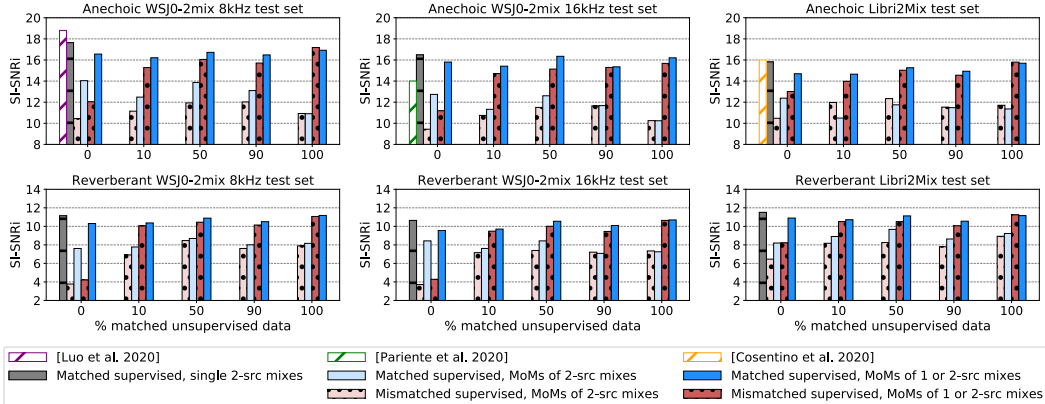

Figure 2: Sweeping proportion of matched unsupervised training examples with matched or mismatched supervised examples on WSJ0-2mix 8kHz (left), WSJ0-2mix 16kHz (middle), and Libri2Mix (right). The leftmost bars in each plot correspond to 100% supervision using PIT, and the rightmost bars are fully unsupervised using MixIT. Pooled within-condition standard deviations are around 2 dB (see Appendix F).

data, while performance is more comparable on reverberant data. However, even a small amount of supervision (10% supervised) dramatically improves separation performance on anechoic data. When the supervised data is mismatched, adding only a small amount of unsupervised data (10%) from a matched domain drastically improves performance: using mismatched anechoic supervised data and matched reverberant unsupervised data, we observe boosts of 2-3 dB for MoMs of 2-source mixtures on all datasets. Training with mismatched supervised and matched unsupervised MoMs of 1-to-2-source mixtures, Si-SNRi increases by about 6 dB on WSJ0-2mix and 2.5 dB for Libri2Mix.

Though our goal is to explore MixIT for less supervised learning, our models are competitive with state-of-the-art approaches that do not exploit additional information such as speaker identity. Figure 2 includes the best reported numbers for 8 kHz WSJ0-2mix [28], 16 kHz WSJ0-2mix [34], and Libri2Mix [9]. We also train 4-output supervised models with PIT on matched single two-source mixtures created dynamically during training (gray bars in Figure 2, see Appendix D for ablations). For WSJ0-2mix 8 kHz, other recent supervised approaches further improve over [28] by incorporating additional speaker identity information, including [32] with 20.1 dB and WaveSplit [51] with 20.4 dB. Note that MixIT is compatible with any of these network architectures, and could also be used in combination with auxiliary information.

## 4.2 Speech enhancement

MixIT can also be useful for tasks where sources are drawn from different classes. An example of such a task is speech enhancement, where the goal is to remove all nonspeech sounds from a mixture. We prepared a speech enhancement dataset using speech from LibriSpeech [33] and non-speech sounds from `freesound.org`. Based on user tags, we filtered out synthetic sounds (e.g. synthesizers). We also used a sound classification model [15] to avoid clips likely containing speech.

Using the clean speech data and filtered nonspeech data, we construct two splits of data: speech-plus-noise audio and noise-only audio. These two types of audio simulate annotating the speech activity of a large amount of data, which can easily be done automatically with commonly-available speech detection models. To ensure that the speech signal is always output as the first separated source, we added constraints to the possible mixings in MixIT. The first mixture $x_1$ is always drawn from the speech-plus-noise split, and the second mixture $x_2$ is always drawn from the noise-only split. A three-output separation model is trained, where the optimization over the mixing matrix $\mathbf{A}$ (3) is constrained such that only outputs 1 and 3 or 1 and 2 can be used to reconstruct the speech-plus-noise mixture $x_1$, and only separated sources 2 or 3 can be used to match the noise-only mixture $x_2$.

As a baseline, we also trained a supervised two-output separation model with our signal-level loss (2) on both separated speech and noise outputs. On a held out test set, the supervised model achieves 15.0 dB SI-SNRi for speech, and the unsupervised MixIT model achieves 11.4 dB SI-SNRi. Thus, by only using labels about speech presence, which are easy to automatically annotate or, unlike ground-truth

Table 1: Multi-source SI-SNR improvement (MSi), single-source SI-SNR (SS), and SI-SNR of reconstructed MoMs (MoMi) in dB for FUSS test set, and MoMi for YFCC100m test set, for different combinations of supervised and unsupervised training data and probability of zeroing out one supervised mixture $p_0$.

| Supervised | $p_0$ | Unsupervised | FUSS | | | YFCC100m |
| | | | MSi | SS | MoMi | MoMi |
|---|---|---|---|---|---|---|
| FUSS (16 hr) | 0.2 | YFCC100m (1600 hr) | **13.3** | 35.4 | 7.7 | 6.1 |
| FUSS (16 hr) | 0.0 | YFCC100m (1600 hr) | 12.5 | 11.1 | 7.7 | 5.9 |
| FUSS (16 hr) | 0.2 | – | **13.3** | **37.1** | 7.7 | 0.3 |
| FUSS (16 hr) | 0.0 | – | 12.4 | 11.9 | 7.4 | 0.2 |
| FUSS (8 hr) | 0.2 | FUSS (8 hr) | 13.2 | 31.4 | 8.3 | 1.3 |
| FUSS (8 hr) | 0.0 | FUSS (8 hr) | 12.9 | 11.4 | 8.5 | 0.9 |
| – | – | FUSS (16 hr) | 12.1 | 4.0 | **10.9** | 4.9 |
| – | – | YFCC100m (1600 hr) | 11.1 | 6.9 | 9.6 | **8.4** |

source waveforms, possible for humans to annotate, we can train a speech enhancement model only from mixtures with MixIT that achieves 76% of the performance of a fully-supervised model. Such a fully-supervised model is potentially vulnerable to domain mismatch, and also requires substantial effort to construct a large synthetic training set. In contrast, MixIT can easily leverage vast amounts of unsupervised noisy data matched to real-world use cases, which we will explore in future work.

## 4.3 Universal sound separation

Universal sound separation is the task of separating arbitrary sounds from an acoustic mixture [21, 43]. For our experiments, we use the recently released Free Universal Sound Separation (FUSS) dataset [47, 48],which consists of sounds drawn from `freesound.org`. Using labels from a prerelease of FSD50k [11], gathered through the Freesound Annotator [12], source clips have been screened such that they likely only contain a single sound class. The 10 second mixtures contain one to four sources, and we construct about 55 hours of training mixtures from about 16 hours of isolated sources.

We also use the audio from YFCC100m videos [40] as a source of a large amount of in-the-wild audio consisting of mixtures of arbitrary sounds. We use a particular subset that is CC-BY licensed and extract ten second clips. After splitting based on uploader, the resulting training set consists of about 1600 hours of audio. These clips are used as unsupervised mixtures to create MoMs.

Table 1 shows separation performance results on FUSS and YFCC100m under a variety of supervised, semi-supervised and purely unsupervised settings. Since the YFCC100m data is unsupervised and thus does not have reference sources, we cannot compute MSi and SS measures. As a surrogate measure, we report the SI-SNRi of using sources separated from a MoM to reconstruct the reference mixtures using the optimal mixing matrix (3), a measure which we call MoMi. We have found that MoMi is generally correlated with MSi (see Appendix G), and thus provides some indication of how well a model is able to separate multiple sources. We also experiment with randomly zeroing out one of the supervised mixtures and its associated sources in the MoM with probability $p_0$ during training. When one of these mixtures is zeroed out, the training example is equivalent to a single supervised mixture, rather than a supervised MoM. In particular, we found that using $p_0 = 0.2$ helps all semi-supervised and fully supervised models greatly improve the SS score, though sometimes with a slight loss of MSi. This means that these models are better at reconstructing single-source inputs without over-separating, with slight degradation in separating mixtures with multiple sources.

From Table 1, for the FUSS test set, the best MSi (13.3 dB) and SS (37.1 dB) scores are achieved by the purely supervised FUSS model with $p_0 = 0.2$, but this model achieves nearly the worst MoMi performance on YFCC100m (0.3 dB). Adding unsupervised raw audio from YFCC100m into training with MixIT achieves comparable MSi, SS, and MoMi (13.3 dB, 35.4 dB ≈ 37.1 dB, 7.7 dB), while greatly improving the MoMi score on YFCC100m (6.1 dB > 0.3 dB). Training a totally unsupervised model on YFCC100m achieves the best overall MoMi scores of 8.4 dB on YFCC100m and 9.6 dB on FUSS, with degraded performance on individual FUSS sources, especially single sources (11.1 dB MSi, 6.9 dB SS). SS scores are lower for purely unsupervised models and supervised models with $p_0 = 0$ since they are never provided with supervised single-source inputs.

Qualitatively, we find that the totally unsupervised model trained with MixIT on YFCC100m can be quite effective at separating in-the-wild mixtures, as described in Appendix H.

## 5 Discussion

The experiments show that MixIT works well for speech separation, speech enhancement, and universal sound separation. In the speech separation experiments, unsupervised domain adaptation always helps: matched fully unsupervised training is always better than mismatched fully supervised training, often by a significant margin. To the best of our knowledge, this is the first single-channel purely unsupervised method which obtains comparable performance to state-of-the-art fully-supervised approaches on sound separation tasks. For universal sound separation and speech enhancement, the unsupervised training does not help as much, presumably because the synthetic test sets are well-matched to the supervised training domain. However, for universal sound separation, unsupervised training does seem to help slightly with generalization to the test set, relative to the supervised-only training, which tends to do better on the validation set. In the fully unsupervised case, performance on speech enhancement and FUSS is not at supervised levels as it is in the speech separation experiments, but the performance it achieves with no supervision remains unprecedented. Unsupervised performance is at its worst in the single-source mixture case of the FUSS task. This may be because MixIT does not discourage further separation of single sources. We have done some initial experiments with possible approaches. One approach we tried was to impose an additional "separation consistency" loss that ensures that sources separated from a MoM are similar to those separated from the individual mixtures (including single-source mixtures). We also tried adding a covariance loss that penalizes correlation between the estimated sources. Both of these approaches improved performance some degree (in particular, improving reconstruction of single sources), but the most effective approach we have found so far is including some amount of matched supervised data. We intend to continue exploring approaches for preventing over-separation in future work.

In some of the experiments reported here, the data preparation has some limitations. The WSJ0-2mix and FUSS data have the property that each unique source may be repeated across multiple mixture examples, whereas Libri2Mix and YFCC100m both contain unique sources in every mixture. Such re-use of source signals is not a problem for ordinary supervised separation, but in the context of MixIT, there is a possibility that the model may abuse this redundancy. In particular in the 1-or-2 source case, this raises the chance that each source appears as a reference, which could make the unsupervised training act more like supervised training. However, the unsupervised performance on Libri2Mix, which does not contain redundant sources, parallels the WSJ0-2mix results and shows that if there is a redundancy loophole to be exploited in some cases, it is not needed for good performance.

Another caveat is that the MixIT method is best suited to cases where sources occur together independently, so that the network cannot determine which sources in the MoM came from the same reference. The fact that unsupervised training works less well on the YFCC100m data, when evaluated on the artificially mixed FUSS data, may be partly because of the non-uniform co-occurrance statistics in YFCC100m, for two reasons. First, the evaluation is done on FUSS, which has uniform co-occurrence statistics, so there is a potential mismatch. Second, in cases where there are strong co-occurrence dependencies, the network could avoid separating the sources within each component mixture, treating them as a single source. It is a future area of research to understand when this occurs and what the remedies may be. A first step may be to impose a non-uniform co-occurrence distribution in synthetic data to understand how the method behaves as the strength of the co-occurrence is varied. An ultimate goal is to evaluate separation on real mixture data; however, this remains challenging because of the lack of ground truth. As a proxy, future experiments may use recognition or human listening as a measure of separation, depending on the application.

## 6 Conclusion

We have presented MixIT, a new paradigm for training sound separation models in a completely unsupervised manner where ground-truth source references are not required. Across speech separation, speech enhancement, and universal sound separation tasks, we demonstrated that MixIT can approach the performance of supervised PIT, and is especially helpful in a semi-supervised setup to adapt to mismatched domains. More broadly, MixIT opens new lines of research where massive amounts of previously untapped in-the-wild data can be leveraged to train sound separation systems.

## Broader impact

Unsupervised training as proposed in this paper has the potential to make recent advancements in supervised source separation more broadly useful by enabling training on large amounts of real world mixture data which are better matched to realistic application scenarios. These systems have many potential benefits as components in assistive technologies like hearing aids or as front ends to improve speech recognition performance in the presence of noise. Preliminary experiments in this work focused on artificial mixtures from standard datasets. However, the ability to leverage unlabeled in-the-wild data, which is comparatively easy to collect, increases the importance of careful dataset curation, lest bias in the data lead to decreased performance for underrepresented groups, e.g. accented or atypical speech. As described above, care must be taken to avoid unintentional correlation between sources, e.g. such that examples containing a particular source class do not only appear with the same backgrounds.

## Acknowledgments and Disclosure of Funding

The idea of using mixtures of mixtures came about during the 2015 JHU CLSP Jelinek Summer Workshop at University of Washington. Special thanks goes to Jonathan Le Roux for prior discussions of other methods using MoMs, and to Aren Jansen for helpful comments on the manuscript.

All authors were employed by Google for the duration of the work described in this paper. There are no other funding sources or competing interests.

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
