[Supplementary Material]

# A   Separation model architecture

In Table 2, we describe the separation network architecture using a TDCN++ [21]. As compared to the original Conv-TasNet method [29], the changes to the model include the following:

- Instead of global layer norm, which averages statistics over frames and channels, the TDCN++ uses instance norm, also known as feature-wise global layer norm [21]. This mean-and-variance normalization is performed separately for each convolution channel across frames, with trainable scalar bias and scale parameters.

- The second difference is skip-residual connections from the outputs of earlier residual blocks to form the inputs of the later residual blocks. A skip-residual connection includes a transformation in the form of a dense layer with bias of the block outputs and all paths from residual connections are summed with the regular block input coming from the previous block. Note that all dense layers in the model include bias terms.

- Finally, a scalar scale parameter is applied after each dense layer stage, which is an over-parametrization trick that improves convergence. The scale parameters for the second dense layer in layer $i$ are initialized using exponential decay in the form of $0.9^i$. All other scales are initialized to 1.0. This initial scaling controls the contribution of each block into the residual sum. It also causes the initial blocks train faster and the later blocks to train slower, which is reminiscent of layer-wise training.

Table 2: Separation network with TDCN++ architecture configuration. Variables are number of encoder basis coefficients $N = 256$, encoder basis kernel size $L$, which is 40 for 16 kHz data and 20 for 8 kHz data, number of waveform samples $T$, number of coefficient frames $F$, and number of separated sources $M$.

| Module name | Operation | Output shape | Kernel size | Dilation | Stride |
|---|---|---|---|---|---|
| Waveform | Input | $T \times 1$ | – | – | – |
| Encoder | Conv | $F \times N$ | $1 \times L \times N$ | 1 | $L/2$ |
| Coeffs | Intermediate | $F \times N$ | – | – | – |
| Initial bottleneck | ReLU | $F \times N$ | – | – | – |
| | Dense | $F \times 256$ | $N \times 256$ | 1 | 1 |
| $i$-th separable dilated conv block (x32) | Input | $F \times 256$ | Previous block output + sum of skip-residual inputs | | |
| | Dense | $F \times 512$ | $256 \times 512$ | – | – |
| with skip-residual | Scale | $F \times 512$ | $1 \times 1$ | – | – |
| connections b/w blocks: | PReLU | $F \times 512$ | – | – | – |
| $i \to i + 1$, | Instance norm | $F \times 512$ | – | – | – |
| $0 \to 8, 0 \to 16, 0 \to 24$, | Depthwise conv | $F \times 512$ | $512 \times 3 \times 1$ | $2^{\mathrm{mod}(i,8)}$ | 1 |
| $8 \to 16, 8 \to 24$, | PReLU | $F \times 512$ | – | – | – |
| $16 \to 24$, | Instance norm | $F \times 512$ | – | – | – |
| | Dense | $F \times 256$ | $512 \times 256$ | – | – |
| | Scale | $F \times 512$ | $1 \times 1$ | – | – |
| Final bottleneck | Dense | $F \times 256$ | $512 \times 256$ | – | – |
| Perform masking | Dense | $F \times M \cdot N$ | $256 \times M \cdot N$ | – | – |
| | Sigmoid | $F \times M \cdot N$ | – | – | – |
| | Reshape | $F \times M \times N$ | – | – | – |
| | Multiply | $F \times M \times N$ | Multiply with $F \times 1 \times N$ coeffs | | |
| Decoder | Transposed conv | $T \times M$ | $L \times N \times 1$ | 1 | $L/2$ |
| Separated waveforms | Output | $T \times M$ | – | – | – |

As mentioned in the text, we also apply a mixture consistency projection [49] to the resulting separated waveforms, which projects them such that they sum up to the original mixture. This projection solves the following optimization problem to find mixture consistency separated sources $\hat{s}$ given initial separated sources $\underline{s}$ separated by the model from a mixture $x$:

$$
\begin{aligned}
& \underset{\hat{\mathbf{s}} \in \mathbb{R}^{M \times T}}{\text{minimize}} && \frac{1}{2} \sum_m \|\hat{s}_m - \underline{s}_m\|^2 \\
& \text{subject to} && \sum_m \hat{s}_m = x.
\end{aligned}
\tag{5}
$$

The projection operation is the closed-form solution of this problem:

$$\hat{s}_m = \underline{s}_m + \frac{1}{M}(x - \sum_{m'} \underline{s}_{m'}), \tag{6}$$

which is differentiable and can simply be applied as a final layer to the initial separated sources $\underline{s}$.

## B  Training details

For each task, we train all models to 200k steps, evaluating a checkpoint every 10 minutes. For evaluation on the test set, we select the checkpoint with the highest validation score. As mentioned in the text, all models are trained with batch size 256 with learning rate $10^{-3}$ on 4 Google Cloud TPUs (16 chips).

## C  Ablations

In order to evaluate the contribution of different components of the proposed model we compare several variations trained on WSJ0-2mix with two-source mixtures: disabling mixture consistency, and varying $\text{SNR}_{\max}$. Performance is reported on the validation set after 200k training steps.

**Mixture consistency** We observed modest improvement of 0.5 dB SI-SNRi by incorporating mixture consistency (6) versus not.

**SNR threshold** Performance is not very sensitive to $\text{SNR}_{\max}$ as long as it is 20 dB or larger, as shown in Table 3.

Table 3: SI-SNRi in dB as a function of $\text{SNR}_{\max}$ for unsupervised MixIT on WSJ0-2mix 2-source mixtures.

| $\text{SNR}_{\max}$ | 10 | 20 | 30 | 40 | 50 |
|---|---|---|---|---|---|
| SI-SNRi | 13.1 | 13.8 | 13.7 | 13.6 | 13.7 |

**Zero source loss** For speech separation tasks using 1-to-2-source mixtures and for universal sound separation on FUSS, the separation model needs to be able to output near-zero signals for "inactive" source slots. Following the implementation of the baseline FUSS separation model [47], we experimented with using explicit losses on separated signals that align to all-zeros reference sources for supervised training examples. Following the baseline FUSS implementation, we chose to use the prescribed modification of the negative SNR loss function (2), where the mixture signal $x$ instead of the source signal $y$ is used to determine the soft-thresholding, where we still set $\tau$ corresponding to $\text{SNR}_{\max}$ of 30 dB:

$$\mathcal{L}_0(y = \mathbf{0}, \hat{y}, x) = 10 \log_{10}\left(\|\hat{y}\|^2 + \tau\|x\|^2\right), \tag{7}$$

which means the loss will be clipped when the power of the separated signal drops 30 dB below the power of the mixture signal. We also experimented with changing the value of $p_0$, the probability of zeroing out one of the mixtures and its corresponding reference signals for supervised examples.

Table 4: Effect of incorporating the zero loss $\mathcal{L}_0$ (7) on supervised separation performance on the WSJ0-2mix and FUSS validation sets without additional reverb after 200k training steps.

| Dataset | $p_0$ | $\mathcal{L}_0$ | SI-SNRi | MSi | SS |
|---|---|---|---|---|---|
| WSJ0-2mix 2-source mixtures | 0.0 | ✗ | 15.9 | | |
| | 0.0 | ✓ | 14.3 | | |
| FUSS | 0.0 | ✗ | | 12.3 | 10.9 |
| | 0.0 | ✓ | | 12.0 | 12.0 |
| | 0.2 | ✗ | | 12.6 | 29.5 |
| | 0.2 | ✓ | | 12.3 | 34.7 |

The results are shown in Table 4 for WSJ0-2mix and FUSS, where the models are trained on mixtures of mixtures, and evaluated on single mixtures from the validation set. Notice that incorporating $\mathcal{L}_0$ decreases SI-SNRi on WSJ0-2mix and MSi on FUSS; however, it boosts SS performance, which is probably due to better suppression of inactive source power. Because we also use mixture consistency, reduction in inactive source power should allow the network to allocate more power to the reconstructed single source. For FUSS, using a $p_0$ greater than 0 slightly improves MSi, and greatly improves SS. This is because the network is presented with actual single mixtures during training, which improves the match between train and test.

# D    Supervised PIT baseline for speech separation

To more fairly compare to supervised PIT speech separation models from the literature [28, 34, 9], we train our separation network in the same way on single two-source mixtures created from the anechoic and reverberant versions of each of the three speech separation datasets. We use either $M = 2$ or $M = 4$ outputs for the separation model. The $M = 4$ outputs are a useful comparison for the MoM-trained speech separation models in Section 4.1. We also experiment with dynamic remixing, which is essentially an augmentation approach that continuously creates new mixtures during training, instead of using static mixtures defined by the datasets.

The results on each of the speech separation test sets are shown in Table 5. Notice that using $M = 4$ outputs, as we also do for MoM-trained models in Section 4.1, only slightly degrades performance compared to using $M = 2$ outputs. Also, dynamic remixing tends to boost SI-SNRi performance by 1 dB or more on WSJ0-2mix, but does not yield significant improvements on Libri2Mix. This is due to the training set size of Libri2Mix being an order of magnitude larger than the training set of WSJ0-2mix. Note that training on MoMs implicitly performs dynamic remixing. Given these ablations, we choose $M = 4$ output models with dynamic remixing to use as baselines in Figure 2.

Table 5: Performance of separation model in terms of SI-SNRi trained with supervised PIT on single 2-source mixtures, training and testing on anechoic and reverberant versions.

| Train data | Output sources $M$ | Dynamic remixing | WSJ0-2mix 8 kHz | | WSJ0-2mix 16 kHz | | Libri2Mix 16 kHz | |
|---|---|---|---|---|---|---|---|---|
| | | | Anechoic | Reverb | Anechoic | Reverb | Anechoic | Reverb |
| Anechoic | 2 | ✗ | 16.7 | 3.4 | 15.5 | 2.7 | 15.8 | 7.9 |
| Anechoic | 4 | ✗ | 15.8 | 2.5 | 15.6 | 2.8 | 15.7 | 7.9 |
| Anechoic | 2 | ✓ | 17.7 | 4.5 | 17.0 | 4.6 | 16.0 | 8.4 |
| Anechoic | 4 | ✓ | 17.6 | 4.6 | 16.5 | 4.7 | 15.8 | 8.2 |
| Reverb | 2 | ✗ | 11.4 | 8.9 | 10.9 | 8.7 | 13.9 | 11.2 |
| Reverb | 4 | ✗ | 11.4 | 8.8 | 11.5 | 8.9 | 13.8 | 11.2 |
| Reverb | 2 | ✓ | 14.0 | 11.3 | 13.6 | 11.0 | 13.7 | 11.3 |
| Reverb | 4 | ✓ | 13.8 | 11.1 | 13.2 | 10.6 | 14.0 | 11.5 |

# E    Single-source performance for speech separation models

As shown in Section 4.3, totally unsupervised universal sound separation models trained with MixIT on arbitrary sources tend to over-separate, and thus do not reconstruct single-source inputs very well at their output, as evidenced by the lower SS scores in Table 1. We also evaluated the speech separation models from Section 4.1 on single-source test sets constructed by simply using the first source in each supervised test mixture as input to the model, and measuring SS (i.e. absolute SI-SNR of the best-matching separated sources). The results are shown in Figure 3. Notice that both supervised and unsupervised models perform poorly when trained on 2-source mixtures. However, training separation models with mixtures that contain 1-or-2-sources dramatically increases the separation performance up to a level where they perform comparably with purely supervised approaches trained with clean sources and PIT.

Figure 3: Same models and conditions as depicted in Figure 2, except the models are presented with a single speech source at their input, and the metric is SS: absolute SI-SNR of best-matching separated source.

# F    Statistical analysis of speech separation results

To visualize the statistical reliability of the results, we compute a normalized standard deviation score that corresponds to the within-condition errors computed in a one-way analysis of variance (ANOVA), as proposed in [27]. For score data $r_{i,j}$, for mixture example $i$, processed with model condition $j$, we compute normalized data:

$$\tilde{r}_{i,j} = r_{i,j} - \mu_j + \mu, \tag{8}$$

where $\mu_j = \frac{1}{N} \sum_i r_{i,j}$ is the mean across conditions for each example, and $\mu = \frac{1}{NJ} \sum_{i,j} r_{i,j}$ is the global mean score. Using the normalized scores, and pooling across conditions, we report a within-condition standard deviation of 2.1 dB for WSJ0-2mix 8 kHz, 1.9 dB for WJ0-2mix 16 kHz, and 2.1 dB for Libri2Mix, roughly corresponding to 95% confidence intervals of 4.1 dB, 3.7 dB, and 4.1 dB, respectively.

# G    Correlation between MoMi and MSi

To measure the correlation between MoMi and MSi, we took models from Table 1 and evaluated them on the test set of FUSS MoMs (1000 examples). Then, for each example, we measured mean MoMi and mean MSi, which are shown in the scatter plots in figure 4. Note that MoMi and MSi are generally correlated, with Pearson and Spearman correlation coefficients of 0.66 and 0.65 for models that use supervised data, and 0.52 and 0.51 for unsupervised models. The lesser degree of correlation for unsupervised models is likely due to these models being more prone to over-separation.

Figure 4: Scatter plots of MoMi versus MSi on FUSS test MoMs for supervised, semi-supervised, and unsupervised models. Note that MoMi and MSi are moderately correlated.

# H    YFCC100m separation examples

Figures 5 and 6 show two randomly-selected example mixtures with separated outputs from an unsupervised MixIT model trained on YFCC100m, corresponding to the model in the bottom row of Table 1. See https://universal-sound-separation.github.io/unsupervised_sound_separation/universal_sound_separation/audio_demos.html (examples 2 and 5 for Figures 5 and 6, respectively) to listen to the corresponding audio and for additional non-cherry picked examples.

Figure 5: Example in-the-wild speech mixture from YFCC100m (top) and the 8 output sources (middle, bottom) from an unsupervised separation model trained on YFCC100m using MixIT. The mixture contains overlapping speech sources, which are reasonably separated by speaker identity. For example, notice the lower pitched voice which only appears in the second half of the clip (middle row, left), is distinct from the higher pitched speaker primarily active between about 2 and 6 seconds (middle right, center left). Also apparent is source consistent primarily of footsteps (bottom row, left).

Figure 6: Example in-the-wild music mixture from YFCC100m and separated sources. Despite not being trained explicitly to separate musical sources, the model is able to roughly separate a pop song into semantically meaningful rhythmic and harmonic components, dominated by: (1) the bass drum (middle row, left), (2) higher frequency rhythmic content, snare drum/shakers (bottom row, center left), (3) slowly varying harmony (bottom row, left), and (4) over-separating the vocals across two sources (middle row, center left and far right). In contrast, fully or semi-supervised models trained with FUSS consistently under-separated the mixture, for the most part using only one output source slot.