[Reviews · NeurIPS 2020]

Review 1

Summary and Contributions: This paper introduces a method for training a sound source separation system from mixed signals and without the need for separated source signals at train time, i.e., in an unsupervised way. The idea is simple, generic and can be applied to any kind of neural architecture. Experiments on speech separation, speech enhancement and universal sound separation tasks using different datasets reveal that the method yields adequate results and can even be competitive with state-of-the-art supervised methods in mismatched conditions.

Strengths: To the best of my knowledge, this is the first unsupervised method for single channel source separation that yields competitive results with supervised methods in mismatched condition, which represents an important milestone for the field. The introduction and methodological parts of the paper are sound, clearly written and easy to follow. The simplicity and genericity of the method make it widely applicable, and unlock the possibility of training source separationmethods on massive, in-the-wild datasets.

Weaknesses: The main weakness of the paper is the experiment section 4 which is sometimes handwavy, hard to follow, and misses details, as detailed in the feedback section below.

Correctness: The claims, method and empirical methodology are mostly correct, up to the remarks listed below.

Clarity: The paper is well written and easy to follow except for section 4, as detailed below.

Relation to Prior Work: The relation to prior wor is clearly and correctly discussed in the paper.

Reproducibility: Yes

Additional Feedback: # After reading the other reviews and authors' rebuttal, I stand by my score of 8. All the reviewers were in favor of accepting, I think this is a strong paper whose strengths largely outweight its minor flaws. 1) In Fig. 2, the meanings of "matched" and "mismatched" are not cleary explained. I think mismatched means "trained with Libri2Mix and tested on WSJ0-2mix" and the other way around, but it is never clearly written. 2) Section 4.1.: The experimental design in this section seems inadequate, because the 1+1 source mixture case exactly corresponds to the supervized case. How do the authors distinguish between the "% of unsupervised data" in the training set and "unsupervised 1+1 mixtures" (whatever it means)? 3) L221: "we also consider using supervised data from a mismatched domain" -> which supervised data from which domain? Please detail this. 4) The way the freesound.org dataset crafted by the authors was designed lacks details. WIll it be released for reproducibility? 5) Section 4.3: how was it ensured that the intersection of FUSS and the freesound.org dataset was empty? 6) L274: what is the meaning of this phrase: "randomly zeroing out one of the supervised mixtures with probability p0" ? 7) L278: The terms MSi and SS are not properly explained. In particular, what is the meaning of the "single source mixture case" (L301) ? 8) L303: What is the meaning of "an additional 'separation consistency' loss" ? 9) L315: What is the meaning of "cases where sources occur together independently" ? 10) L326: "this remains challening because of the lack of ground truth": this is not a very good argument because there exists plenty of real datasets where individual sound sources are recorded separately from the same microphone array in the same environment, and can then be easiliy mixed together for testing purpose (e.g. the numerous ChiME challenges). Typos: -L76: improving performance universal sound separation -> improving performance of.. -L86: form -> from.


Review 2

Summary and Contributions: Proposes and demonstrates an effective technique for learning to separating multi-speaker audio by predicting mixtures of mixtures.

Strengths: 1. The idea is relatively simple yet insightful and evidently works well. 2. The problem is well motivated and certainly unsupervised source separation is an important problem. 3. Good empirical demonstrations on a variety of audio separation tasks. Overall, this is a nice paper, a simple yet clever approach to unsupervised source separation with well executed experiments and well written.

Weaknesses: It would be useful for the authors to comment on whether or not this algorithm can be used in tasks other than just audio and if so why not experiment on some to show that it generalizes to mixed images as well.

Correctness: Yes.

Clarity: Nicely written.

Relation to Prior Work: Yes

Reproducibility: Yes

Additional Feedback:


Review 3

Summary and Contributions: This paper proposed an unsupervised method, referred to as remixing and permutation invariant training (RemixPIT), for the sound separation task. The traditional supervised approaches use synthetic mixtures to do the training, which suffers from the big gap between the training data and real data. In RemixPIT, models trained on the mixture of mixtures require to separate them into several sources and remix the sources to approximate the original mixtures. Moreover, RemixPIT can be used with the supervised paradigm parallelly. RemixPIT has the advantage of leveraging the vastly available data in the wild.

Strengths: + This paper proposes a new unsupervised approach for sound separation, which can handle the differences in acoustic conditions and distribution of sources between training and inference process. Such a gap is really a big problem in this research field and RemixPIT provides a potential solution to it. + Extensive experiments on WSJ0-2mix and Libri2Mix demonstrates the effectiveness of RemixPIT. Besides, on the universal sound separation tasks, the proposed method also achieves state-of-the-art performance, especially on the reverberant cases. + RemixPIT has the advantage of leveraging the vastly available data in the wild.

Weaknesses: - The authors claim that synthetic mixture data is problematic and the proposed RemixPIT can tackle this problem. Can the authors provide the separation results on the cases of natural sound mixtures? Since the presented experiment results are still based on the synthetic mixtures, as well as the demo examples in the supplementary. - In part 3.2, the authors assume the number of mixtures to be 2 and the optimization of matrix A is based on the brute-force search. If the number of mixtures goes higher, the search space will exponentially rise. Could the authors present a potential solution for it? - For the speech separation experiments in part 4.1, is the rightmost model trained from scratch on the unsupervised training examples, or based on the model parameters trained with part supervised examples? - From Table 1, the improvement over the pure-supervised learning method is minor, especially for FUSS (16hr).

Correctness: The method is correct, but the supporting experiments for the claims can't be found in the paper. Please refer to the weakness part for the details.

Clarity: Yes.

Relation to Prior Work: Yes.

Reproducibility: No

Additional Feedback: ===== Post Rebuttal ====== It's a nice paper. However, as far as I am concerned, this paper just shows the separation results of simulated data. The experiment will be much stronger with recording results. I understand that it is almost impossible to compute the SNRi of simulated data. But other metrics such as WER can help. Moreover, this paper only has a modest improvement over the purely supervised learning method on FUSS. Combining the opinions above, I decided to give 6 for the final rating.


Review 4

Summary and Contributions: The paper proposes an unsupervised method for sound separation. Unlike supervised systems where sound sources are mixed together to form input and reference output pairs, this paper proposes a method which relies only on the mixtures. The proposed method is applied on speech separation, speech enhancement and sound separation.

Strengths: The proposed method is novel and interesting. It is an unsupervised method which opens up the possibility of training sound separation models directly on real world data. The paper also provides some interesting insight and discussions on the proposed method. Demo examples are also provided.

Weaknesses: I think the paper is a bit weak on the experimental side. All experiments are done on synthetically created training data. This pulls down the paper a bit as the main advantage of this unsupervised training is the ability to use real world data. One of the dataset even has not been released yet. Detailed comments are below.

Correctness: Yes

Clarity: yes

Relation to Prior Work: yes

Reproducibility: Yes

Additional Feedback: Overall, I liked this paper. It is a simple yet clever reformulation of PIT to train sound separation models in an unsupervised manner. The unsupervised training method opens up the possibility of learning from real world data. I also liked the discussion provided in the paper. I think the primary limiting factor of this paper is the experimental evaluation. Given that the focus is on unsupervised learning and developing the ability to use real world data, it would have been more convincing to actually show that empirically. All of the experiments are done on synthetically mixed training data. I think there should have been at least one experiment where RemixPIT training is done using some real world data rather than synthetically mixed ones, and compared with supervised training. How much difference between the two exists ? Matched and mismatched conditions in this case would have been also interesting to see. Some additional comments. Using just upto 2 sources in speech separation seems a bit limiting again. In speech separation experiments (Fig 2), what is a statistically significant change in SN-SNRi ? In Fig 2, in some cases the performance goes up while going from fully supervised on one extreme to fully unsupervised on the other ? While one can argue that this might be expected for mismatched-supervised cases, this also happens for matched-supervised cases, e.g. matched-supervised 2 sources. Can authors comment a bit on it ? For speech enhancement experiments, why draw x1 from speech+noise and x2 from noise only ? WHy not train by just relying on the mixtures, speech+noise set ? It would be better to provide some additional details on the speech enhancement experiments. Is it a matched noise type test or mismatched noise type ? ---------- Overall I think this paper should be accepted. Updating the score to reflect that.

[Author Response · NeurIPS 2020]

Thank you to the reviewers for providing helpful feedback which will improve the paper. We appreciate that the reviewers found our method novel, simple and generic yet effective, and compelling for its potential impact of training sound separation on real-world mixtures. We hope that the specific points below address outstanding issues.

1. *R3, R4: Concern about evaluation, that we train on synthetic mixtures, while the ultimate goal is to use real mixtures.* We agree this is a limitation for the results on some tasks. However, the paper does show results with training on real in-the-wild Freesound mixtures for universal separation (see Sec. 4.3.). Evaluation is another matter: we used synthetic data with source-level references so that we

|  |  |  | FUSS |  |  | YFCC100M |
|---|---|---|---|---|---|---|
| Supervised | $p_0$ | Unsupervised | MSi | SS | MoMi | MoMi |
| FUSS | 0.2 | YFCC100M | 13.6 | 34.6 | **6.8** | 9.0 |
| FUSS | 0.0 | YFCC100M | 12.5 | 11.1 | 6.6 | 9.0 |
| FUSS | 0.2 | – | **13.7** | **35.6** | 6.5 | 4.8 |
| FUSS | 0.0 | – | 12.4 | 11.9 | 6.4 | 4.9 |
| – | – | FUSS | 11.9 | 3.6 | 5.2 | 8.4 |
| – | – | YFCC100M | 10.8 | 6.9 | 4.8 | **10.6** |

can measure objective source-level SNR. Such evaluations are not possible with real-world mixtures, but we plan on adding some surrogate evaluations in the final revision. We plan to separate mixtures of 2 in-the-wild mixtures (MoMs) into their constituent sources, and measuring SI-SNRi of the RemixPIT-reconstructed *mixtures*, called MoMi, which correlates with source-level MSi (see table).

2. *R1: Section 4 is main weakness.* We revised section 4 to be more clear, in line with other specific responses here.

3. *R1 1) Meaning of matched vs mismatched?.* "Matched" means e.g. train on anechoic, test on anechoic, and "mismatched" means e.g. train on anechoic, test on reverberant, and vice versa. We will clarify this in the final revision.

4. *R1 2) Section 4.1.: The 1+1 source mixture case exactly corresponds to the supervised case.* $p\%$ of each training batch are unsupervised mixtures drawn from a $p\%$ subset of a dataset of 1-or-2-source mixtures. The remainder of the batch are supervised mixtures with corresponding reference sources drawn from the subset complement. The model has to infer the number of sources for unsupervised mixtures. Although easier than 2-source MoMs, we think that 1-or-2 sources is more realistic, because single-speaker audio is typical in the real world.

5. *R1 4) Data availability.* We intend to release data recipes for all non-released data, including reverberant version of Libri2Mix, Librivox+Freesound speech enhancement data, and clips used from Freesound and YFCC100M.

6. *R1 5) How was it ensured that FUSS and freesound do not intersect?* There may be some overlap. To avoid this, we reran experiments on independent YFCC100M audio as unsupervised in-the-wild data (see table), with similar results.

7. *R1 6) L274: Meaning of "randomly zeroing out one of the supervised mixtures with probability p0"?* With probability $p_0$, one of the mixtures in the supervised MoM is set to zero, along with its sources, equivalent to using a single mixture.

8. *R1 7) L278: MSi and SS not properly explained.* Sorry for the lack of explanation, we've clarified the text. MSi is SI-SNRi of sources from mixtures containing 2 or more sources, and SS is absolute SI-SNR for single-source mixtures.

9. *R1 10) L326: "remains challenging because of the lack of ground truth"* The reviewer points out a useful method for collecting more realistic supervised data. Unfortunately, it is not ideal as background noises in each recording will be added together in the resulting mixtures. Note that RemixPIT performs similar remixing of recordings except that it also handles separation of background noise as well as overlapping speech in each mixture. So RemixPIT could be applied to CHiME-5, for example, which contains significant non-stationary background noise (the other CHiME datasets are synthetic). However, SNR evaluation without signal-level ground truth would still be a problem with CHiME-5.

10. *R2: Can RemixPIT be used in other domains e.g. images?* Yes, and such experiments are interesting future work. Though visual objects are more commonly simply occluded in reality, transparency is an analogue of audio mixing.

11. *R3: Brute-force RemixPIT implementation won't scale to more mixtures.* As noted in the paper, we intend to defer this to future work. We believe exact solutions have exponential complexity, but good approximations can suffice under certain assumptions. Also note that using more mixtures may increase the mismatch with the unmixed data.

12. *R3: How are rightmost models in section 4.1 plots trained?* From scratch with RemixPIT on unsupervised data only.

13. *R3: From Table 1, the improvement over the pure-supervised learning method is minor.* Good point, we'll be more precise about claims of improvement in revision. Our new evaluations (table above) that show semi-supervised models achieve comparable performance to purely supervised models for FUSS, but also improve MoMi on YFCC100M.

14. *R4: Using just up to 2 sources in speech separation seems a bit limiting again.* We agree, but for the speech separation experiments we chose to focus on standard tasks with ground-truth source references to measure performance.

15. *R4: Statistically significant change in SI-SNRi?* In the final revision, we plan to report this for all experiments.

16. *R4: Performance of fully-supervised speech separation models is sometimes exceeded by fully unsupervised models.* The supervised models in our initial submission were 4-output models trained on supervised MoMs with 2-4 sources, and thus is still a bit mismatched to 2-source test mixtures. However, using MoMs implicitly remixes sources, acting as training data augmentation. This helps on WSJ0-2mix, which is small. Since submission, we reduced this mismatch and improved matched supervised performance by training on individual mixtures instead of MoMs while also using explicit source remixing augmentation. We'll include these results in the final revision.

17. *R4: Questions about speech enhancement.* Training from just mixtures is possible, but we would need another method to determine which estimated source is speech. Light supervision of speech presence and the RemixPIT constraints force the model to always output speech as source 1. Noise is matched (i.e. always drawn from Freesound).

[Meta-Review · NeurIPS 2020]

Reviewers agree that the proposed approach is simple, clever, and significantly novel. Experimental work shows it works well on synthetic data. The only major concern remaining after the rebuttal and discussion phases is the lack of evaluation on real-world data - while being able to leverage real data is claimed as an advantage of the approach. In the end, the consensus after the discussion period was the paper should be accepted, as the proposed idea is interesting enough.